# Social Support, Religious Involvement and Alcohol Use among Students at a Conservative Religious University

**DOI:** 10.3390/bs7020034

**Published:** 2017-05-24

**Authors:** Wendy E. Thompson

**Affiliations:** School of Social Work, College of Public Service, Jackson State University, Jackson, MS 39211, USA; wendy.e.thompson@jsums.edu

**Keywords:** alcohol, social support, religious involvement

## Abstract

The misuse of alcohol among college students remains a significant public health concern in the United States. Excessive drinking among college students has been linked to numerous negative consequences, including rape, impaired academic performance, absenteeism from work and school and damaged social relations. This study examined whether religious involvement and social support played a role in reducing the frequency of alcohol use. A non-random convenience sample of 364 students from a larger study of 760 college students—18 years old and older—were recruited over a 2 month period. The survey used in this study consisted of 124 items and collected information on areas such as substance misuse, sexual activity, use of pornography, relationships, personal religious practices, and social support. A descriptive analysis and chi-square were performed to determine if there was a relationship between frequency of alcohol use and gender, marital status, student class, GPA, religion, ethnicity and age. Linear regression was conducted to determine if social support and religious involvement were predictors of frequency of alcohol misuse. Multivariate regression analysis was used for predicting religious involvement when including social support while controlling for gender, age, ethnicity and grade. The present study revealed that religious involvement was a predictor for reduced frequency of alcohol use, while social support was not a predictor of lower frequency of alcohol use.

## 1. Introduction

The misuse of alcohol among college students remains a significant public health concern in the United States. National data indicate that in 2013, 59.4% of full time college students between the ages 18 and 22 reported drinking in the past month, with 39% engaging in binge drinking in the past month [1]. Around 20% of college students meet the criteria for an Alcohol Use Disorder [2]. Excessive drinking among college students has been linked to numerous negative consequences including rapes, impaired academic performance, absenteeism from work and school and damaged social relations [3].

Despite growing research on the topic and numerous college education and prevention programs to increase awareness on the dangers of alcohol and drugs, the incidence of drinking among college students continues with serious outcomes. For example, each year approximately 696,000 college students are assaulted by another student who has been drinking and 97,000 students experience a sexual assault or date rape [4]. This and other associated outcomes underscore the need for continuous research at understanding the use of alcohol among college students.

Towards this end, several studies have investigated the effects of social support as a protective factor against alcohol misuse. This study builds upon the efforts of Menagi, Harrell, and June [5] by examining these two variables, social support and religious involvement among college students at a conservative Seventh-Day Adventist University and is guided by the hypothesis that religious involvement and social support are associated with reduced rates of alcohol use. As a result of the serious personal, legal, and academic costs associated with the misuse of alcohol, it is important to examine the following research questions:(1)What is the relationship between frequency of alcohol use and gender, marital status, student class, GPA, religion, ethnicity, and age?(2)Are social support and religious involvement predictors of frequency of alcohol use?

## 2. Literature Review

### Social Support, Religious Involvement and Alcohol Use

The transition to life in college can be a perplexing time for many young adults. Thus, perceptions of social support and being cared for by those in one’s life can be very important to levels of wellbeing [6]. Social support has been defined in the literature in numerous ways. In this present study, social support is defined as “any process through which social relationships might promote health and well-being” [7]. This can take the form of instrumental support (i.e., providing a direct service), emotional support (i.e., listening and providing acceptance), or informative support (i.e., providing advice or knowledge) [8]. Social support includes real or perceived resources provided by others that enable a person to feel cared for, valued, and part of a network of communication and mutual obligation [9]. The benefit of supporting students’ well-being has been demonstrated in studies by Maddox and Prinz [10] showing that students with higher levels of emotional functioning earned higher grades, were more academically intelligent, and had greater levels of relational bonding with peers.

Research has identified several risk factors for college alcohol use and has shown them to exist at the individual, family, school and community levels [11]. Individual risk factors include low self-esteem, aggressive and impulsive behavior, poor decision making skills and anger management [12]. However, research has also shown that students who experience protective factors are likely to demonstrate greater resiliency despite their risk exposure to heavy alcohol use. Scales [13] and the National Institute on Drug Abuse [14] identified the protective factors with respect to risk behaviors; the participation in making healthy decisions, involvement in school and community activities, strong sense of religiosity, positive peer relationships and lack of peer approval to alcohol use. Protective factors at the family level include high levels of family connectedness [12] and family support. School and community factors include opportunities for positive involvement and empowerment as well as a sense of belonging, which promotes healthy decision-making among youth and reduces negative externalized behaviors [15,16].

Several studies have suggested that social support may be one mechanism by which the benefits of religious involvement on mental health outcomes are produced [17]. These studies focused on the relationship of religion and social support and found that students with stronger religious commitment, based on frequency and type of religious activities, had a stronger social network of relationships, and scored higher on psychological sense of community [18,19]. Individuals who belong to religious communities perceive greater social support [20,21,22,23,24]. The perceived social support provided by religious involvement has the potential to act as a protective mechanism or buffer against negative health outcomes during stressful events [7]. Menagi et al. [5] examined the mediating role of social support in the relationship between religiousness and alcohol use among college students. This support may act as a buffer against negative health outcomes during stressful life events [7] and may also act as a protective coping mechanism against drinking [25].

Although both religiosity and spirituality can include religion according to the literature, there are conceptual and operational distinctions between the two [26]. Religiosity often refers to both religious behaviors and religious attitudes and encompasses an organized system of beliefs, practices, rituals and symbols designed to facilitate closeness to a higher power, or ultimate truth and foster one’s sense of connection and responsibility to others in the faith community [5]. Both public and private elements of religiousness have been found to be inversely related to alcohol use [27]. This current study focuses only on religiosity in reference to how religious a person is from mainly a social and religious involvement perspective [28].

Conservative religious groups are defined in our article as being Christian and adhering to fundamentalist and /or conservative doctrines. Within these conservative religions, substance use or abuse is seen as pulling the individual away from God and thus any substance use is to be avoided [29]. Research has shown that students in these types of conservative religious subcultural settings, such as Christian college campuses, tend to exhibit less substance use overall than students in other college settings [21,30]. Previous studies by Chitwood et al. [26] and Felt, McBride and Helm [31] have empirically shown that an inverse relationship exists between religiosity and crime, delinquency or problem behaviors like substance use and misuse, with some proposing that religiosity acts as a social control agent, protecting people from engagement in deviance.

## 3. Methods

Data for this study were derived from a survey administered during the months of March to May 2012 at a small, conservative, religiously affiliated university in the Midwest that prohibits alcohol use as a condition of University attendance. Permission to conduct this study was granted by the University’s Institutional Review Board. A cross-sectional study was conducted to collect data using a convenience sample of 760 college students aged 18 years of age and older. This survey was part of an ongoing quinquennial study of risk and protective behaviors on this university campus beginning in 1989, and is the sixth such study [32]. Two survey instruments were given due to the large number of variables and limited time frame for completing the survey. 364 students from the larger study of 760 students completed survey questions that pertained to alcohol use, religious involvement and social support.

Although a convenience sample was used in this study, precautions were taken to make this sample representative of the social demographic characteristics of the university. First, class periods were chosen based on the large number of classes offered at that time and the large number of students likely to be taking those classes, as well as the likelihood that those classes would contain a broad range of socio-demographic characteristics overall university population. The university is committed to providing a drug-free environment for learning and working. Such a commitment led the University to establish a Drug-Free Policy, which outlines clearly the University’s zero-tolerance position and strives to educate the student body on the advantages of a drug-free lifestyle. Students are expected to remain drug-free. Drug-free means abstaining from the use of alcohol, tobacco and other mind-altering drugs. It also means refraining from the misuse and/or abuse of prescription drugs. The University also upholds all laws which prohibit the possession, use, manufacturing or distribution of controlled substances [33].

From the pool of classes meeting at the times chosen, classes which consisted of a substantial number of undergraduate and graduate students were selected for inclusion. Professors of these classes were contacted for permission to survey their class. Questionnaires were administered during classes, over the course of one class period, by trained proctors. Students were instructed not to provide any identifying information on their questionnaire, and were assured that participation was voluntary and all information would be kept confidential.

## 4. Survey Measures

The questionnaire, a broad health risk and behaviors survey, was designed with numerous sections to concentrate on areas such as substance misuse, sexual activity, use of pornography, relationships, personal religious practices, and social support. The survey consisted of 124 questions and was administered using a paper and pencil close-ended question format.

### 4.1. Dependent Variables

The dependent variable is alcohol use and is defined as the frequency of alcohol use within the year prior to the administration of the questionnaire: “Within the last year about how often have you used alcohol”. The question included seven response categories (never; tried it once; once/month; twice/month; once/week; 3 times/week; daily). Prior year use was chosen since it would relate most closely with the relational independent variables.

### 4.2. Independent Variables

Independent variables included religious involvement and social support. Christian Religious Internalization Scale (CRIS) (Department of Psychology, New York, NY, USA) is a validated scale developed by Ryan et al. (1993), that was used to measure religious involvement. Religious involvement was defined as participation in both public and private displays of religiosity. The survey question asked “How often do you participate in the following activities listed below”, with public displays measured through: “Attend church services”; “Attend Sabbath School” (similar to Sunday School); and, “Attend school-sponsored religious programs”. Private displays were measured through: “Personal prayer”; “Read the Bible (outside of class assignments); “Read Seventh-Day Adventist literature outside of class assignments”; and “Family worship”. Responses were marked on a nine-point scale (never; less than once a year; about once or twice a year; several times a year; about once a month; two to three times a month; nearly every week; every week; and, several times a week).

### 4.3. The Multidimensional Scale of Perceived Social Support (PSSS)

The Multidimensional Scale of Perceived Social Support (PSSS) (Department of Pediatrics, Indianapolis School of Medicine, IN, USA) is a validated 12-item instrument designed to assess perceptions about support from family, friends and significant others. The scale was developed by Zimet et al. (1988). The items are divided into factor groups relating to the source of support, with choices ranging from ‘a lot, some, a little and none’. Sample items included “How many close friends do you have (people you feel at ease with, can talk to about private matters, and can call on for help” and “How many faculty/staff you can discuss sensitive issues like drugs/sex with?)”. Cronbach’s Alpha on the items in the PSSS = 0.80.

## 5. Statistical Analysis

Returned questionnaire data were examined for response biases or within- survey inconsistencies. Questionnaires containing inconsistent or suspicious response patterns were removed from the data set. The data were analyzed using (SPSS) version 20.0 (IBM, Armonk, NY, USA). Descriptive statistics were conducted to determine frequency of alcohol use, social support, and demographics.

Correlations were also used to gauge the relationship between self-reported substance use religious involvement and social support. Linear regressions were used to predict which independent variable—social support or religious involvement—would predict frequency of alcohol use. Three levels of analyses were employed in summarizing and describing the data. Univariate analysis was carried out to describe the demographic characteristics of the sample. The Bivariate analysis was done to gauge the relationship between each control and independent variables. Multivariate analysis was conducted to determine which independent variable/s (social support/religious involvement) were predictors of frequency of alcohol use.

## 6. Results

The present study is based on an analyses of data collected from 334 college students enrolled at one university. As shown in Table 1, more than half the students (60.2%) were female and single (82.7%). Nearly one-third (30.3%) were sophomores and juniors and almost half (49.2%) of the students had a B average grade. Seventh-Day Adventist was the dominant religion (94.4%), followed by those without a religious preference. White/non- Hispanics comprised the largest ethnic group (45.1%), followed by African Americans (19.4%), Latino (16.1%), Asian/Pacific Islander (13.5%), and West Indian (5.9%). Ages ranged from 18 to 63 years. Most participants were young adults (M = 21.79, SD = 5.55) and the majority (61.9%) were aged 18–22 years. The socio-demographic characteristics of the sample approximated those of the wider student population. For instance, the race/ethnic composition of the university population included Whites (37.0%), African Americans (18.3%), Asian/Asian American (13.2%), Latino/a (14.3%), and other groups (11.2%). At the time of the study, the student population was 59.1% female, and 90.2% of students identified as Seventh-Day Adventist. Of interest, the majority of students (51.5%) lived on campus, and those who wished to consume alcohol would normally car pool to a setting where alcohol was consumed. The nearest liquor store was 20 miles from the university.

Please see Frequency of Alcohol Users and Nonusers by Demographics in Table 2.

What is the relationship between frequency of alcohol use and gender, marital status, student class, GPA, religion, ethnicity and age?

A chi-square test of independence was used to see if there was a relationship between various categorical variables. This test examined the relationship between frequency of alcohol use and gender, marital status, student class, GPA, religion, ethnicity and age. The relationship between frequency of alcohol use and gender was not statistically significant, X^2^ (3, N = 334) = 6.05, *p* = 0.109. The relationship between frequency of alcohol use and marital status was not statistically significant, X^2^ (3, N = 334) = 6.78, *p* = 0.079.

Participants’ reported use of alcohol did not differ by class standing; freshman, sophomore/junior, senior and grad/professional, X^2^ (9, N = 334) = 6.27, *p* = 0.712. The relationship between the frequency of alcohol use and GPA was statistically significant, X^2^ (5, N = 334) = 14.4, *p* = 0.026. In the religious category, the relationship between the frequency of alcohol use and religion was statistically significant, X^2^ (6, N = 334) = 23.8, *p* = 0.001, however, some cells are extremely small and as a consequence, the statistical test might not be reliable. For the ethnicity category, there was not a statistically significant relationship between ethnic groups and frequency of alcohol use, X^2^ (12, N = 334) =8.99, *p* = 0.703. The probability level might have been influenced by the way the variable was measured and the distribution of responses across cells, for example, just having 3 categories for ethnicity (white, black and other)

In the age category, there was a significant relationship between frequency of alcohol use and age X^2^ (3, N = 334) = 14.2, *p* = 0.003).

Table 3 presents the frequencies and percentages of responses to each item on the two scales which measured the independent variables. When students were asked “How many close friends do you have (people you feel at ease with, can talk to about private matters, and can call on for help?)” almost one-fourth (23.8%) indicated that they had at least three to five friends compared with those who said none (1.4%). When asked “How many faculty/staff can you discuss sensitive issues like alcohol/drugs/sex with?” almost one third (31.8%) of the students indicated none compared to 12.9% who said one or two. When asked, “How many relatives do you have that you feel close to?” 17.4% shared that they had three to five relatives that they feel close to. When asked “How many of these friends or relatives do you see at least once per month?” 17.1% indicated that they see one or two relatives at least once monthly compared to those who said none (4.2%). When asked “Do you belong to any social, recreational, work, church or other community groups? (For example, social clubs, exercise groups, campus ministries or community service)?” almost one third (31.9%) indicated that they belonged to one to five groups. Finally, when students were asked “How many close friends do you have (people you feel at ease with, can talk to about private matters, and can call on for help)?” about (23.8%) indicated that they had at least three to five friends compared with those who said none (1.4%).

Table 4 presents the responses to the questions relating to religious involvement: namely, how often do you participate in activities such as; attending church services, personal prayer, read the Bible (outside of class assignments), family worship, attend Sabbath School, read Seventh-Day Adventist literature outside of class assignments and attend school-sponsored religious programs. Almost three quarters of the respondents (71.0%) indicated that they attend church nearly every week, every week or several times per week as compared to 1.8% who said “never.” Almost four-fifths (79.3%) indicated that they prayer nearly every week, weekly or several times per week as compared to 1.5% who said “never”. Almost one third (60%) indicated that they read their Bible nearly every week, weekly or several times per week as compared to 2.8% who said “never”. Almost one third (30.3%) indicated that they have family worship nearly every week, weekly or several times per week as compared to (21.7%) who said “never.” Almost one third (32.8%) indicated that they attend Sabbath School several times a year, about once a month or 2 or 3 times a month as compared to 21% who said “never.” 44.7% indicated that they attend school-sponsored religious programs nearly every week, weekly or several times per week as compared to 10.1% who said “never.” Just over a third (34.4%) indicated that they read SDA literature several times a year, about once a month or at least two to three times a month compared with 20.7% who said “never.

Are social support and religious involvement predictors of frequency of alcohol use?

Table 5 shows that Linear Regression was calculated to predict alcohol use based on two independent variables; social support and religious involvement. The model summary gave a measure of how well the overall model fits, and how well the predictor variables (social support and religious involvement) predicted alcohol use. *R* is a measure of how well our predictions predict the outcome, but the square of *R* was taken to get a more accurate measure. *R*^2^ gives the amount of variance in frequency of alcohol use explained by social support and religious involvement. A significant equation was found (F = 2333) = 33.464, *p* < 0.000 with an *R*^2^ of 0.167. Therefore, we can say that 16.7% of the variance in the dependent variable can be explained by the two independent variables. The analysis shows that social support did not significantly predict alcohol use, (ß = 0.027, t (335) = 0.536, *p* < 0.05), however, religious involvement did significantly predict alcohol use, (ß = −0.413, t (335) = −8.115, *p* > 0.05). These findings do not imply causality, because this was a cross-sectional study and not a longitudinal study.

In Table 6, we conducted multivariate analyses with four variables (Gender, Age, Ethnicity and GPA). We controlled for these four variables while looking at religious involvement and social support to find out how each affects alcohol use individually. The following was found: gender was a strong predictor of frequency of alcohol use and this was statistically significant (*p*-value = 0.013, 0.001 respectively). Examination of students between the ages of 18–22 and students 23 and above, showed that both were statistically significant (*p* = 0.000, 0.000 respectively), and were strong predictors of frequency of alcohol use. When comparing ethnic groups Whites, Hispanics and other ethnic groups (*p* < 0.001) used alcohol less because of their religious involvement. African American (*p* > 0.05), almost reached significance, but were the only group that was not a predictor of frequency of alcohol use. Another predictor of less frequency of alcohol use was GPA (*p* value = 0.000).

## 7. Discussion

The purpose of this study was to examine whether religious involvement and social support played a role in reducing the frequency of alcohol use. The results of this study did support past research findings that consistently suggested that religious involvement is a protective factor against alcohol use in adults, and is related to lower frequency of alcohol use [34,35]. In one study, both religiousness dimensions and emotional social support were related to less frequent alcohol use, but mediation was not supported. This present study revealed that only religious involvement was a predictor for reduced frequency of alcohol use, and social support was not a predictor of lower frequency of alcohol use.

Based on the findings of this study, a major protective element in keeping college students from drinking alcohol is the strength of their religious involvement. At least in this study, social support did not provide a protective effect on avoiding alcohol consumption. These findings indicate that it is relevant to assess the degree to which religious practices and beliefs saturate various life purviews.

We must interpret these findings with caution because the sample was limited to students at one religiously conservative school (Seventh-Day Adventist). Given that, there are very few religious traditions that reject all alcohol use for their members it is difficult to extrapolate the role that protective role that religion plays in other teetotalling traditions. However, the present study does provide meaningful firsthand evidence that religious involvement (distinct from religiosity) may actually decrease the frequency of alcohol use within some populations.

The current study has other important limitations as well. First, the sample consisted of 45.1% Caucasian students as compared to 19.4% African Americans, and still lower percentages for other ethnic groups. Perhaps the role of religious involvement and its connection to social support may be different for ethnic minority groups, particularly those that adhere to non-Western religion. Future research on the relationship between alcohol use and religious involvement should consider the role of ethnicity and non-Western religions.

Alcohol normally plays a role in building social support networks in college settings. Because drinking behaviors are normative to college life [36] social support from friends and colleagues in some cases may take the form of drinking in a group. Thus, social support may enhance drinking behaviors during this critical transition into adulthood. However, in this study, social support was not associated with either increased or decreased drinking behaviors Instead, religious involvement had an independent and primary effect, perhaps relating to the strength of the internalized beliefs around abstaining from alcohol that are deeply engrained into Seventh-Day Adventist church members.

One of the reasons for conducting this study was to identify risk and protective factors for alcohol use among one conservative, non-alcohol-using university. Future studies should include other teetotalling colleges and should further investigate different motivating factors for drinking as related to religious involvement and drinking behaviors.

Finally, although the hypothesis that religious involvement and social support would be related to alcohol use was not supported, this hypothesis might be more applicable in other less conservative populations or participants with a history of heavy alcohol use patterns.

## 8. Conclusions

One of the main reasons for engaging in this study was to examine whether religious involvement and social support played a role in reducing the frequency of alcohol use. Results from this study revealed that only religious involvement was a predictor for reduced frequency of alcohol use, and social support was not a predictor of lower frequency of alcohol use. Based on the findings of this study, a major protective element in deterring college students from drinking alcohol is the strength of their religious involvement, while social support did not provide a protective effect on avoiding alcohol consumption. The bond to a religious persuasion and and/or belief systems seem to act as the glue to a set of social values, symbols, behaviors and practices, i.e., the adhesion to a comprehensive and complex religious practices, which includes, among other things, the acceptance or refusal of alcohol and drugs.

## Figures and Tables

**Table 1 behavsci-07-00034-t001:** Demographic Characteristics of Participants Surveyed (N = 334).

		No	%
**Gender**	Male	133	39.8
	Female	212	60.2
**Marital Status**	Single	258	82.7
	Married	54	17.3
**Class Status**	Freshman	70	21.2
	Sophomore/Junior	100	30.3
	Senior+	81	24.5
	Grad/professional	79	23.9
**GPA/Grade**	A/A+	144	43.2
	B/B+	164	49.2
	C+ and below	25	7.5
**Religion**	Seventh-Day Adventist	303	94.4
	Other Protestant	8	2.5
	None	10	3.1
**Ethnicity**	African American	59	19.4
	White	137	45.1
	Asian/Pacific	41	13.5
	Latino	49	16.1
	West Indian	18	5.9
**Age**	18–22	190	61.9
	23 and above	117	38.1

**Table 2 behavsci-07-00034-t002:** Analysis of Frequency (%) of Alcohol Users and Nonusers by Demographics (N = 334).

	Never	Tried Once	Once Month/2 Month	Once Week/Daily	Number	X^2^	*p* Value
**Gender**							
Male	78.9	6.0	8.3	6.8	133		
Female	68.7	9.5	15.9	6.0	201	6.05	0.109
**Marital Status**							
Single	70.5	8.1	14.0	7.4	258		
Married	87.0	5.6	3.7	3.7	54	6.78	0.079
**Student Class**							
Freshman	71.4	10.0	14.3	4.3	70		
Sophomore/Junior	71.0	9.0	12.0	8.0	100		
Senior	67.9	9.9	16.0	6.2	81		
Graduate	81.0	3.8	8.9	6.3	79	6.27	0.712
**GPA/Grades**							
A/A-	81.3	6.3	9.0	3.4	144		
B/B-	64.6	11.0	16.5	7.9	164		
C/C-D/F	76.0	0.0	12.0	12.0	25	14.3	0.026
**Religion**							
None	30.0	10.0	40.0	20.0	10		
Seventh-Day Adventist	75.6	8.3	11.2	5.0	303		
Other Protestant	37.5	0.0	37.5	25.0	8	23.8	0.001
**Ethnicity**							
African American	76.3	10.2	11.9	1.7	59		
White/non-Hispanic	70.1	5.8	14.6	9.5	137		
Asian/Pacific Islander	70.7	12.2	14.6	2.4	41		
Latino	71.4	8.2	12.2	8.2	49		
West Indian	72.2	11.1	5.6	11.1	18	8.99	0.703
**Age**							
18–22	67.9	12.6	12.6	6.8	190		
23+	80.3	0.9	13.7	5.1	117	14.2	0.003

**Table 3 behavsci-07-00034-t003:** Independent Variables: Social Support.

Questions	Responses	N	%
How many close friends can you talk to about to about private matters?	None	11	1.4
	1or 2	101	13.3
	3 to 5	181	23.8
	6 to 10 or more	71	9.4
How many Faculty/Staff can you discuss sex, alcohol, and drugs with?	None	242	31.8
	1or 2	98	12.9
	3 to 5	15	2
	6 to 10 or more	8	0.11
How many relatives you feel close to?	None	32	4.2
	1or 2	114	15
	3 to 5	132	17.4
	6 to 10 or more	88	10.5
How many close friends/relatives you see at least once monthly?	None	73	9.6
	1or 2	130	17.1
	3 to 5	96	12.6
	6 to 10 or more	64	8.4
What is the total number of groups to which you belong?	None	78	10.3
	1or 2	141	18.6
	3 to 5	101	13.3
	6 to 10 or more	26	0.34

**Table 4 behavsci-07-00034-t004:** Independent Variables: Religious Involvement.

Questions	Responses	N	%
How often do you attend church services?	Never	7	1.8
	Less than once a year	3	0.8
	Several times a year About once a month/2/3 times a month/	64	16.2
	Nearly every week/every week/several times per week	281	71.0
How often do you have personal prayer?	Never	6	1.5
	Less than once a year	3	0.8
	Several times a year About once a month/2/3 times a month/	31	7.8
	Nearly every week/every week/several times per week	314	79.3
How often do you read your Bible?	Never	11	2.8
	Less than once a year	15	3.8
	Several times a year About once a month/2/3 times a month/	86	21.7
	Nearly every week/every week/several times per week	236	59.6
How often do you have family worship?	Never	86	21.7
	Less than once a year	20	5.1
	Several times a year About once a month/2/3 times a month/	114	28.8
	Nearly every week/every week/several times per week	120	30.3
How often do you attend Sabbath School?	Never	83	21.0
	Less than once a year	17	4.3
	Several times a year About once a month/2/3 times a month/	139	32.8
	Nearly every week/every week/several times per week	117	29.5
How often do you attend school religious programs?	Never	40	10.1
	Less than once a year	18	4.5
	Several times a year About once a month/2/3 times a month/	110	27.8
	Nearly every week/every week/several times per week	177	44.7
How often do you read SDA literature?	Never	82	20.7
	Less than once a year	25	6.3
	Several times a year About once a month/2/3 times a month/	136	34.4
	Nearly every week/every week/several times per week	103	26.0

**Table 5 behavsci-07-00034-t005:** Linear Regression Predicting Alcohol Use Based on Social Support and Religious Involvement.

Variable	B	Se	*p*-Value
Social support	0.061	0.113	0.592
Religious involvement	−1.156	0.143	0.000
Constant	4.732		
*R*^2^	0.17		

**Table 6 behavsci-07-00034-t006:** Multivariate Analysis for predicting Religious Involvement when including social support while controlling for gender, age, ethnicity and grade.

Variables Religious Involvement		*R*^2^	B	β	T	Sig (*p* Value)
**Gender**	Males	0.045	−0.674	−0.218	−2.505	0.013
	Females	0.207	−1.244	−0.462	−7.225	0.001
**Age**	18–22	0.105	−0.872	−0.328	−4.703	0.000
	23+	0.190	−1.332	−0.450	−5.187	0.000
**Ethnicity**	African American	0.029	−0.432	−0.179	−1.345	0.184
	White	0.204	−1.432	−0.459	−5.890	0.000
	Latino	0.303	−2.065	−0.553	−4.583	0.000
**GPA/Grade**	A	0.257	−1.424	−0.527	−7.035	0.000
	B	0.118	−0.958	−0.345	−4.665	0.000

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
