# Peer review of "Social Support, Religious Involvement and Alcohol Use among Students at a Conservative Religious University"

_behavsci, 2017, doi:10.3390/bs7020034_

Round 1

Reviewer 1 Report

Title: Social support, religious involvement and alcohol use among students at a conservative religious university

Reviewer’s comments:

1. It is probably best to omit the name of the university (in the abstract, on page 10 line 234, and in Table 3).  Just describe it as (for example) a medium-sized (private?) university located in a particular region of the United States. 

2. Page 5, lines 134-135.  “The instrument that included the variables of interest was completed by 364 students.”  There needs to be a bit of clarity here.  In the same paragraph, the author notes that 760 college students participated in the study.  Why didn’t all the 760 students complete the survey that included the “variables of interest”?  Why were there two surveys?  How was it decided which students completed which survey?  Confusing.  If I am interpreting correctly, the entire analyses in the present study was conducted with 364 cases.  If I am correct, then the author might state something like, “364 students from the larger study of 760 students completed survey questions that pertained to alcohol use, (etc).”  In the abstract, it suggests that the present paper is focusing on the 760 students.  That will need to be changed, assuming that the data for this article pertain to 364 students. 

3. Bottom of page 5.  Can the author provide a comparison of some socio-demographic data of the sample and of the overall student population for that year?    For example, what is the percentage of females and blacks at the university in the year that the data were collected?  Compare those data to %female and %Black in the sample, and then comment on representation.  The author is aware that generalizing to a wider population is not possible here.  Still, it would be good to see whether the sample demographics approximate those of the wider student population. 

4. Is there any way of knowing whether the majority of students lived on campus?  Also, do students who live on campus tend to have easy access to transportation to/from campus?  Are alcohol establishments and stores within walking distance of the campus?  What are the penalties when students are caught drinking alcohol on campus? This kind of material could be summarized in one paragraph. It would help set the context for the reader. 

5. Dependent variable – please provide some comment on the limitations of using such an extensive time frame (i.e., one year) in the measure for alcohol use.   For example, I am wondering whether “heavier” drinkers might have a more difficult or an easier time trying to recall the frequency of alcohol consumption over the past 12 months. 

6. Table 2 – race/ethnicity results.  My guess is that the relationship just might reach statistical significance if the variable was measured differently, e.g., white, black, other.  Perhaps state that the probability level might have been influenced by the way the variable was measured and the distribution of responses across cells?

7. Table 2 – I agree that the bivariate relationship between religion and alcohol use appears to be statistically significant, however, there are some cells that are extremely small (e.g., 10 students or fewer) and as a consequence, the statistical test might not be reliable.  The author should mention this possibility in the paragraph on page 10.   

8. The proportions of White and Black students listed on page 16 (bottom) are different than the proportions listed earlier in the paper.  Perhaps the earlier percentages were based on the full sample of 760 students?

9. Tables 5, 6, & 7 can be combined into one table.  Avoid copying and pasting from the SPSS tables directly into the paper.  I strongly suggest that the author construct his/her own table from the results in the SPSS output.  There are several good examples in the literature.  First, however, there are a few things that might need revised with regard to the analytic strategy (see below).    

10. It seems like the multivariate analysis has got to include a control for age.  Can the author insert that variable into the equation?  Actually, there’s room for a few additional independent variables.  Some variables might not be statistically significant in bivariate analyses but can emerge as significant in multivariate analyses.  I would keep it simple.  Retain the two independent variables of theoretical interest (social support and religious involvement), add an independent variable for race (perhaps recode into dummy variables for white, black, and other), gender (for example code females as 1; males =0. I presume that data on trans were not collected at a religious-oriented university).  Add a binary variable for age.  The author has already coded Age into two categories.  I would revise based on drinking age (I do recognize that loads of young people consume alcohol regardless of age):  21 and older=coded 1; 20 and younger=coded 0.   

11. The author should mention that s/he is running multiple regression with a dependent variable with 4 categories.  Yes, other authors have used the same strategy for 4-category dependent variables although some authors would insist on say, 6 categories or more with multiple regression.  There’s an alternative although I am not certain whether the author wants to explore it.  Given that the vast majority of students had NOT consumed alcohol during the past year, why not create an alternative dependent variable and code it 1 for abstained, and 0 for not abstained.  Then use the logistic regression strategy in SPSS with that dependent variable.  Logistic regression is geared for DVs with two categories only.  Use the exact same independent variables, and include controls for say, age, race, and gender.  Results might be very interesting. 

Minor editing: 

12. Page 15, line 286.  I would begin this section with a statement such as:  “Multivariate regression results are presented in Table 5.” [and then just have the one table for the multivariate results]. 

13. Page 2, line 52.  I prefer the word “misuse” rather than “abuse.”  The latter sounds somewhat judgmental.  See also page 4, line 114.

14. Page 3, line 69.  Replace “advantage” with the word “benefit.”

15. Page 3, line 79.  “…(2005) identified protective factors with respect to risk behaviors; participation…”  Perhaps revise to “…(2005) identified the protective factors with respect to risk behaviors: participation”  (inserted the word “the” and replaced the semi-colon with a colon).

16. Page 4, lines 105-106.  “…designed to facilitate closeness to God, higher power, or ultimate truth…”  I understand the meaning here.  The journal has an international readership, many of whom stem from diverse religious and spiritual backgrounds. Thus, it is probably more acceptable to revise the sentence as such:   “…designed to facilitate closeness to a higher power, or ultimate truth…”  In comparison, it is appropriate to use the term “God” in the bottom paragraph because the author mentions that the paper focuses on conservative Christian groups. 

17. Page 7, line 186.  “gage” should read “gauge.”

18. Page 7, line 192.  “Analysis” should read “analysis” (i.e., lower case).

19. Page 22, line 471.  Should read:  McBride.  See also page 5, line 132. 

20. Page 22, line 471.  Should read: Terry-McElrath. 

21. The citation and reference style will be need to be revised if the article is accepted. 

Thank you.

Author Response

I have made the necessary corrections and have attached a file

Dr. Wendy Thompson 

Reviewer 2 Report

The investigation of the relationship between college student level of alcohol consumption and two independent variables, religious involvement and social support has been examined by others, as your lit. review indicates. The examination of this question within a sample of students who attend a fundamentalist Christian university has not been widely studied and can make a contribution to the field.

The following are recommendations for strengthening your investigation/paper.  Overall, the paper would be strengthened by tightening up the focus of the study as presented in the narrative.  As it now stands, it almost seems that parts of the content of the narrative could have been taken from a larger study and some elements of that study have found their way into the narrative which causes some confusion for this reader.  Specific suggestions appear below:

The explanation of the sample size & instrumentation is confusing. The abstract says the N=760, but the narrative says N=364. Two questionnaires are mentioned, but only one is used.  A clearer explanation is needed about how the final sample was reached and whether there are methodological biases related to this smaller sample n.

Is there a reason why the variable, social support, is not included in the list of variables in the abstract?

Multiple risk factors e.g., self esteem and protective factors e.g., family connectiveness are discussed in the paper but are not in the study/analysis. A more focused presentation of the specific independent factors examined in the study, without inclusion of  other factors that are not analyzed, would help the reader understand where the paper is going.  Furthermore, the mediating role of social support in the relationship between religiousness and alcohol use is discussed, but the analysis is not designed to assess this relationship.

The components of the PSSS and the religiousness scale are presented in tabular form in Tables 3 & 4, but the reason for including this information needs to be explained.

The Chi Square presentation in Table 2 needs attention.  For example, I assume only one statistical value was calculated for Student Class, but the table contains 4 identical values.  This is confusing.  Also, some of the variables e.g., Religion, contain very small cells which may distort the meaning of the statistic.

Multiple control variables are presented in bivariate format e.g., age, but none seem to be included in the regression analysis.  The reason for their exclusion needs to be explained.  Is there a reason why there are no control variables in the regression models.

You may have something of a unique opportunity here:  Have you considered an analysis that only includes the students that belong to the Seventh Day Adventist student body?  By eliminating the relatively few other respondents, you would be in a position to examine your research question about social support, religious behavior, and alcohol use within a fundamentalist/conservative Christian group.  Findings from such an analysis could make a unique contribution to the field.

Author Response

I have attached a file with corrections.

Dr. Thompson

Round 2

Reviewer 1 Report

Please see attached comments.

Author Response

I would like to thank you for reading my article and giving me candid feedback, it was very helpful.

See attachment with corrections   

Reviewer 2 Report

This draft demonstrates improvements in the description of the nature of the study, variables studied, and sample size and certain additional aspects of the study design. 

The analysis, however, has now become a demographic-specific analysis i.e., separate models have been run for Gender-specific [one for males & one for females]  and additional demographic-specific analyses. This is a method of eliminating the influence of a third variable and is useful in its own right.  But this is not the same as controlling simultaneously for all demographic variables in one regression model.  These results are interesting and meaningful in their own right, even though none of these demographic-specific models appears to control for the additional demographic variables.  If this has not been done,  the discussion should reflect that this is a series of demographic-specific models and that results may be limited by the fact that in each model, no other demographic have been controlled during a specific analysis.  That is to say, you do not know how the results might be different if those controls were in place.

I recommend a little more caution in statements about investigating whether religious involvement and social support "play a role in reducing" alcohol use.  This is a cross sectional study and by definition the study is looking at associations whose directional relationship is not as clear as it would be in a longitudinal examination of these relationships.  You do have interesting findings, but I recommend caution so as not to overstate your findings.

There needs to be more improvement in the discussion on p 6 about whether the sample is reflective of the demographics of the total university population.  I recommend you include in your table of demographics the demographics of the total population in the semester the data were collected.  This would make it clear how similar the sample and the population data are.

On p 6, line 144 there is a confusing phrase about "...African American females 18.8% compared to 41.1% Whites".  This needs to be rewritten or removed.  Lines 145-146 also add confusion in that the sum of the % living on and % living off campus adds up to less than 92%.  Are there really a third group of students who live in an third situation that is neither on nor off campus?

The way you construct the measure of religious involvement still is not discussed adequately.  This needs to be done and related to the literature: Is your measure a summed scale or index?  Is the measure a replication measure [if so references are needed] or a new measure?  And if new in particular, has any effort been made to demonstrate statistically that it has methodological rigor, and how is it related to the literature on religiosity?  This issue is continued in line 351 where the paper states that this measure is not a measure of religiosity.  There are, however,  multiple papers that indicate that religious involvement is a dimension of religiosity, which is a multidimensional concept.  You will strengthen the paper if you tie your work more closely into the existing data on religiosity and drug and alcohol use.  Your bibliography actually contains more than one article that does  that.

I am puzzled as to why there are multiple identical Chi-Square values within your crosstabs or demographics and other variables.  I assume you conducted only one chi-square test per crosstab.  If so, only one chi-square value is usually presented  along with one p-value for each of those cross-tab tables.

In your limitations section, I recommend that you include the fact that yours is a cross-sectional analysis which limits your ability to draw conclusions about directionality.  Again it is a matter of making cautious conclusions that do not overstate what can be concluded from a cross-sectional design.

Author Response

Thank you for giving valid feedback to my article, I really appreciated the suggestions.

See attached corrections.

Round 3

Reviewer 1 Report

File attached.

Author Response

Reviewer#1: Thank you for taking the time to read my article and supplying me with the necessary feedback to improve my work.

Manuscript ID: behavsci-174312
Type of manuscript: Article
Title: Social Support, Religious Involvement and Alcohol Use Among
Students at a Conservative Religious University
Authors: Wendy Thompson *
Emails: 
[email protected]

Comments from reviewer

1.   Results section, lines 188-203

I appreciate that the author included the socio-demographic profile of the university students as a whole and compared those data with the sample characteristics. I suggest that the author revise this section so that it flows a bit better. Perhaps revise to something like this:

The present study is based on an analyses of data collected from 334 college students enrolled at one university. As shown in Table 1, more than half the students (60.2 %) were female and single (82.7%). Nearly one-third (30.3%) were sophomores and juniors and almost half (49.2%) of the students had a B average grade. Seventh-Day Adventist was the dominant religion (94.4%), followed by those without a religious preference. White/non- Hispanics comprised the largest ethnic group (45.1%), followed by African Americans (19.4%), Latino (16.1%), Asian/Pacific Islander (13.5%), and West Indian (5.9%). Ages ranged from 18 to 63 years. Most participants were young adults (M = 21.79, SD = 5.55) and the majority (61.9%) were aged 18-22 years. The socio-demographic characteristics of the sample approximated those of the wider student population. For instance, the race/ethnic composition of the university population included Whites (37.0%), African Americans (18.3%), Asian/Asian American (13.2%), Latino/a (14.3%), and other groups (11.2%). At the time of the study, the student population was 59.1% female, and  90.2% of students identified as Seventh-Day Adventist. Of interest, the majority of students (51.5%) lived on campus, and those who wished to consume alcohol would normally car pool to a setting where alcohol was consumed. The nearest liquor store was 20 miles from the university.

Corrected by revising lines 188-203 as indicated by the reviewer.

2.   Table 5

I do not know whether the journal will publish a table with an SPSS table, that is, a copy/paste of SPSS results. Most of the time, we create our own tables based on the results in the SPSS output. I would think that Table 5 would only need results for the following:

Variable                                             B            se                 p-value

Social support                                .061      .113                  .592

Religious involvement              -1.156      .143                  .000

Constant                                       4.732

R2                                                    .17

Corrected by revising Table 5 to   reflect reviewers comments

3.   Table 6

OK it’s really good to include some of the demographic variables in the model. Age and ethnicity are measured with dummy variables so one of those categories has to be omitted from the model and then the omitted category becomes the reference category.  I am assuming that this is a regression model?

Corrected, yes, this is regression model.
